Revisiting the type material of two African Diplozoinae (Diplozoidae: Monogenea), with remarks on morphology, systematics and diplozoid specificity

http://orcid.org/0000-0002-2641-9059 Dos Santos Quinton Marco
http://orcid.org/0000-0001-8820-7679 Avenant-Oldewage Annemariè aoldewage@uj.ac.za
Department of Zoology, Faculty of Science, University of Johannesburg (Auckland Park Campus) , Johannesburg, Gauteng , South Africa
Justine Jean-Lou
Electronic publication date: 2024 Feb 28
Publication date: 2024
Volume: 12
Electronic Location ID: e17020
Received 2023 Aug 24; Accepted 2024 Feb 6
Copyright: © 2024 Dos Santos and Avenant-Oldewage
Copyright year: 2024
Copyright holder: Dos Santos and Avenant-Oldewage
License: This is an open access article distributed under the terms of the Creative Commons Attribution License, which permits unrestricted use, distribution, reproduction and adaptation in any medium and for any purpose provided that it is properly attributed. For attribution, the original author(s), title, publication source (PeerJ) and either DOI or URL of the article must be cited.
License URL: https://creativecommons.org/licenses/by/4.0/

Keywords: Platyhelminthes, Fish parasite, Biodiversity, Freshwater, Paradiplozoon, Polyopisthocotyla

Funding: Oppenheimer Memorial Trust (2022) University of Johannesburg (UJ) Global Excellence and Stature (2018–2021) National Research Foundation (NRF) Parasitological Society of Southern Africa UJ (FRC, URC) NRF 116067 Annemariè Avenant-Oldewage’s research trust fund This work was supported by the Oppenheimer Memorial Trust (2022) and the University of Johannesburg (UJ) Global Excellence and Stature (2018–2021) for post-doctoral fellowships, and the National Research Foundation (NRF) for a doctoral scholarship, to Quinton Marco Dos Santos. The Parasitological Society of Southern Africa provided the WO Neitz Memorial Scholarship to QMDS and Annemariè Avenant-Oldewage’s research trust fund, funded the study of specimens at the SNMNH. Annemariè Avenant-Oldewage received grants from UJ (FRC, URC) and the NRF (GRANT 116067) for running expenses. Annemariè Avenant-Oldewage’s research trust fund co-funded the visit to the SNMNH and to top up funding for five consecutive post-doctoral fellowships to Quinton Marco Dos Santos. The funders had no role in study design, data collection and analysis, decision to publish, or preparation of the manuscript.

==============================
The morphological characterisation of Diplozoidae spp. is highly reliant on the details of the sclerotised components of the hooks and clamps in the haptor. Only six species of Paradiplozoon (Diplozoinae) have been described from Africa, four of which have adequate morphological and even comparative ITS2 rDNA data available. However, the descriptions of Paradiplozoon ghanense (Thomas, 1957) and Paradiplozoon aegyptense (Fischthal & Kuntz, 1963) lack essential taxonomic information, specifically the details for their haptoral sclerites. As such, all available material from museum collections for these two species were studied using light microscopy to supplement the original morphometric descriptions. The holotype and paratypes of P. aegyptense were studied, but only voucher material for P. ghanense could be sourced. However, this voucher material for P. ghanense was deposited by the species authority and bore a striking resemblance to the illustrations and collection details from the original description. They were thus identified as the type series for the taxon, with a lectotype and paralectotype designated. Both P. ghanense and P. aegyptense could be readily distinguished from other taxa based on the supplementary data generated here, supporting their distinctness. The haptoral sclerites of P. aegyptense were most similar to those of Paradiplozoon krugerense Dos Santos & Avenant-Oldewage, 2016, also described from Labeo spp., while the sclerites of P. ghanense were most similar to Paradiplozoon bingolense Civáňová, Koyun & Koubková, 2013 and Paradiplozoon iraqense Al-Nasiri & Balbuena, 2016. Additionally, a voucher of P. aegyptense collected from the alestid type host of P. ghanense was reidentified as the latter species here. This greatly simplified the known host specificity for Paradiplozoon spp. in Africa, with P. aegyptense now exclusively reported from Cypriniformes (Cyprinidae and Danionidae), and P. ghanense restricted to Characiformes (Alestidae). The occurrence of all diplozoids from non-cyprinoid hosts was also investigated and several records of diplozoids occurring on non-cyprinoid hosts were collated and scrutinised. Excluding the two instances of diplozoids described and exclusively occurring on Characiformes fishes (P. ghanense and Paradiplozoon tetragonopterini (Sterba, 1957)), most other non-cyprinoid collections appear sporadic and unsubstantiated, but warrant further investigation supported by diligent taxonomic data. Even though the morphometric descriptions of both P. ghanense and P. aegyptense were fully reported on here, additional material will be needed to study their genetic profiles and phylogeny.

Introduction

The Diplozoidae Palombi, 1949 are hermaphroditic parasites which mature and reproduce after two larval diporpa fuse in permanent cross-copula on a suitable host. They attach to the gills of their teleost fish hosts using a haptor armed with clamps and a pair of hooks. The sclerotised structures making up these clamps and hooks are of great taxonomic importance, being crucial for diplozoid species differentiation. The morphology of the anterior end of the median sclerite and the sclerites connecting this structure to the clamp jaws are of particular importance in this regard, as well as the size of the hooks. The size of the clamps has also been used for diplozoid species identification historically, but the size of these structures can be a function of other variables such as host size and environmental factors, making it an unreliable taxonomic feature (Gläser & Gläser, 1964; Matějusová et al., 2002; Milne & Avenant-Oldewage, 2012). Within the Diplozoidae, there are two subfamilies differentiated by the number of clamp pairs in the haptor. Diplozoinae Palombi, 1949 have four pairs of clamps and contain five genera, while Neodiplozoinae Khotenovsky, 1980 have more than eight pairs of clamps and have two monotypic genera. All Diplozoinae species in Africa currently belong to the genus Paradiplozoon Akhmerov, 1974, while the Neodiplozoinae are represented by a single Afrodiplozoon Khotenovsky, 1981 species.

Of the six Paradiplozoon spp. in Africa (Table 1), Paradiplozoon ichthyoxanthon Avenant-Oldewage in Avenant-Oldewage, Le Roux, Mashego & Jansen van Vuuren, 2014, Paradiplozoon vaalense Dos Santos, Jansen van Vuuren & Avenant-Oldewage, 2015, Paradiplozoon krugerense Dos Santos & Avenant-Oldewage, 2016 and Paradiplozoon moroccoense Koubková, Benovics & Šimková in Benovics, Koubková, Civáňová, Rahmouni, Čermáková & Šimková, 2021 were described relatively recently. Adequate taxonomic morphometric data is available for these species either from their original descriptions, or subsequent sclerite studies (Dos Santos & Avenant-Oldewage, 2015; Dos Santos, Dzika & Avenant-Oldewage, 2019), allowing sufficient morphological and morphometric comparison to other diplozoid taxa. Data for the second internal transcribed spacer of ribosomal DNA (ITS2 rDNA) is also available for these four species, enabling effective comparison with other diplozoids. In contrast, the descriptions of the remaining two species, Paradiplozoon ghanense (Thomas, 1957) and Paradiplozoon aegyptense (Fischthal & Kuntz, 1963), lack key taxonomic characteristics for proper taxonomic differentiation. Unidentified diplozoid parasites have also been reported in Africa by Thurston (1970) and Truter et al. (2023), but these species were not fully identified or described.

Table 1 Collection records of Diplozoinae from Africa and African diplozoids from other localities.

Species	Recorded as	Host	Recoded as	Locality	Record	
Paradiplozoon aegyptense
(Fischthal & Kuntz, 1963)	Diplozoon aegyptensis	Brycinus macrolepidotus
Valenciennes, 1850	Alestes macrolepidotus	Butiaba, Lake Albert, Uganda	Paperna (1979) *	
Enteromius paludinosus
(Peters, 1852)**	Barbus paludinosus	Nzoia River, Kenya	Paperna (1979)	
Petr & Paperna (1979)	
Labeo coubie Rüppell, 1832	Labeo cubie	Lake Volta, Black and White Volta Confluence, Ghana	Paperna (1969)	
Lake Volta, Ghana	Paperna (1979)	
Labeo cylindricus Peters, 1852	–	Ruaha River, Tanzania	Paperna (1979)	
Labeo forskalii Rüppell, 1835	–	Giza Fish Market, Cairo, Egypt	Fischthal & Kuntz (1963)	
Lake Albert, Uganda
Egypt	Paperna (1979)	
Labeo sp.	–	Ruaha River, Tanzania	Paperna (1979)	
Labeo victorianus
Boulenger, 1901**	–	Nzoia River, Kenya	Paperna (1979)	
Petr & Paperna (1979)	
Raiamas senegalensis
(Steindachner, 1870)	Barilius loati	Aswa River, White Nile System, Uganda	Paperna (1979)	
Carassius carassius
(Linnaeus, 1758)	–	Kashmir Valley, India	Ahmad et al. (2015a) ***	
Ahmad et al. (2015b) ***	
Cyprinus carpio Linnaeus, 1758	Cyprinus carpio communis	Indus and Suru Rivers, Ladakh, India
Fish farms, Ladakh, India	Dar et al. (2012a) ****	
Indus River, Leh District, Ladakh, India	Dar et al. (2012b) ****	
Schizopyge niger (Heckel, 1838)	Schizothorax niger	Indus and Suru Rivers, Ladakh, India
Fish farms, Ladakh, India	Dar et al. (2012a) ****	
Indus River, Leh District, Ladakh, India	Dar et al. (2012b) ****	
Dal Lake, Kashmir Valley, India	Ahmad et al. (2015a)	
Kashmir Valley, India	Ahmad et al. (2015b) ***	
Schizothorax plagiostomus
Heckel, 1838	Schizothorax plagiostomum	Dal Lake, Kashmir Valley, India	Ahmad et al. (2015c)	
Schizothorax progastus
(McClelland, 1839)	Schizothorax progastrus	Indus River, Leh District, Ladakh, India	Dar et al. (2012b) ****	
Paradiplozoon ghanense
(Thomas, 1957)	Diplozoon ghanense	Alestes baremoze (Joannis, 1835)	Alestes baremose	Yeji, Lake Volta, Ghana	Paperna (1969)	
Brycinus macrolepidotus
Valenciennes, 1850	Alestes macrolepidotus	Lawra, Black Volta Riever, Ghana	Thomas (1957)	
–	Paperna (1979)	
–	Otuocha, Anambra River, Nigeria	Echi & Ezenwaji (2010)	
Paradiplozoon ichthyoxanthon
Avenant-Oldewage in Avenant-Oldewage, le Roux, Mashego & van Vuuren, 2014	–	Labeobarbus aeneus
(Burchell, 1822)	–	Vaal River Barrage, Vaal River, South Africa	Avenant-Oldewage et al. (2014)	
Vaal Dam, Gauteng, South Africa	Avenant-Oldewage & Milne (2014)	
Avenant-Oldewage et al. (2014)	
Gilbert & Avenant-Oldewage (2016a)	
Gilbert & Avenant-Oldewage (2016b)	
Gilbert & Avenant-Oldewage (2016c)	
Gilbert & Avenant-Oldewage (2017)	
Sures et al. (2018)	
Dos Santos, Dzika & Avenant-Oldewage (2019)	
Gilbert, Jirsa & Avenant-Oldewage (2022)	
Below Vaal River Barrage, Vaal River, South Africa	Gilbert & Avenant-Oldewage (2016a)	
Paradiplozoon sp.	Labeobarbus aeneus
(Burchell, 1822)	–	Vaal Dam, Gauteng, South Africa	Milne & Avenant-Oldewage (2006)	
Milne & Avenant-Oldewage (2012)	
–	Labeobarbus kimberleyensis
(Gilchrist & Thompson, 1913)	–	Vaal Dam, Gauteng, South Africa	Avenant-Oldewage & Milne (2014)	
Gilbert et al. (2020)	
Gilbert, Jirsa & Avenant-Oldewage (2022)	
Paradiplozoon sp.	Labeobarbus kimberleyensis
(Gilchrist & Thompson, 1913)	–	Vaal Dam, Gauteng, South Africa	Milne & Avenant-Oldewage (2012)	
Paradiplozoon krugerense
Dos Santos & Avenant-Oldewage 2016	–	Labeo congoro Peters, 1852	–	Olifants River, Kruger National Park, South Africa	Dos Santos & Avenant-Oldewage (2016)	
Labeo rosae Steindachner, 1894	–	Flag Boshielo Dam, Mpumalanga, South Africa	Dos Santos & Avenant-Oldewage (2016)	
Olifants River, Kruger National Park, South Africa	Dos Santos & Avenant-Oldewage (2016)	
Paradiplozoon moroccoense
Koubková, Benovics & Šimková in Benovics, Koubková, Civáňová, Rahmouni, Čermáková & Šimková, 2021	Paradiplozoon moroccoensis	Luciobarbus pallaryi
(Pellegrin, 1919)	Luciobarbus lepineyi	Zouala Oasis, Morocco	Benovics et al. (2021)	
Paradiplozoon vaalense
Dos Santos, Jansen van Vuuren & Avenant-Oldewage 2015	–	Labeo capensis (Smith, 1841)	–	Below Vaal River Barrage, Vaal River, South Africa	Dos Santos & Avenant-Oldewage (2015)	
Labeo umbratus (Smith, 1841)	–	Visgat, Vaal River, South Africa	Dos Santos, Jansen van Vuuren & Avenant-Oldewage (2015)	
Vaal Dam, Gauteng, South Africa	Dos Santos, Jansen van Vuuren & Avenant-Oldewage (2015)	
Dos Santos & Avenant-Oldewage (2015)	
Dos Santos, Dzika & Avenant-Oldewage (2019)	
Paradiplozoon sp.	–	Cheilobarbus serra (Peters, 1864)		Olifants-Doorn River system, Western Cape, South Africa	Truter et al. (2023)	
Sedercypris calidus (Barnard, 1938)		Olifants-Doorn River system, Western Cape, South Africa	Truter et al. (2023)	
Diplozoon sp.		Alestes sp.		Butiaba, Lake Albert, Uganda	Thurston (1970) *****	
Notes:

* Paradiplozoon ghanense based on the findings presented here.

** Only diporpa juveniles collected.

*** Details of which host and parasites were collected from specific sites not provided, but Wular Lake; Anchar Lake; Dal Lake; Manasbal Lake; River Jhelum and River Sindh are mentioned.

**** Unclear if all host species were infected with P. aegyptense, hosts listed only.

***** Resembled P. ghanense.

Both P. ghanense and P. aegyptense were described as Diplozoon von Nordmann, 1832, but subsequently moved to Paradiplozoon when the family was revised by Khotenovsky (1985). As the text is in Russian and not freely accessible, later works still consider these taxa as Diplozoon species (Echi & Ezenwaji, 2010; Dar et al., 2012a, 2012b; Ahmad et al., 2015a, 2015b, 2015c). Paradiplozoon ghanense was described from Brycinus macrolepidotus Valenciennes, 1850 (Alestidae) from the Black Volta near Lawra, Ghana. This was the first diplozoid description from Africa, and one of only two diplozoid species to be described from non-cyprinoid host species. The initial description provides most morphometric data required for diplozoid taxonomy, except for sclerite detail of the attachment clamps and the size and shape of the central hooks. According to Thomas (1957), no hooks were detected, and he noted that “it is probable that they were torn out during removal of the parasite from the gills” (sic.). The description of this species included two diagrams: one illustrating a complete specimen with internal organs and the overall body shape, and the second presenting a clamp. The clamp illustration unfortunately lacked detail, appeared stylised, and is thus not taxonomically informative. Paradiplozoon aegyptense was described from Labeo forskalii Rüppell, 1835 (Cyprinidae) obtained from the Giza fish Market in Egypt (Fischthal & Kuntz, 1963). Like P. ghanense, the description of P. aegyptense provided most morphometric data and included a detailed diagram illustrating a whole parasite with the arrangement of internal organs. Unfortunately, no diagrams of the haptoral sclerite details were provided, although the description of P. aegyptense did include measurements for the central hook. Moreover, the description of the clamp sclerites of P. aegyptense was exceptionally brief, only stating that they are similar to those of P. ghanense.

Various attempts to obtain fresh material for both P. ghanense and P. aegyptense were unsuccessful and thus only museum deposited material could be studied. Most of the type material (holotype and seven paratypes) for P. aegyptense was deposited to the U.S. National Museum Helminthological Collection (later the U.S. National Parasite Collection) and is now housed at the Smithsonian National Museum of Natural History (SNMNH), Washington. Material for P. aegyptense (one paratype and two vouchers) was also found in the collection of the Royal Museum for Central Africa (RMCA), Tervuren, Belgium. The type material of P. ghanense was not designated nor the collection indicated by Thomas (1957). However, two voucher specimens deposited by the species authority are housed at the SNMNH. With the exception of the data generated by Sicard et al. (2003) for an unidentified diplozoid from a recorded host of P. aegyptense in the Ivory Coast, and unsubstantiated and unlikely data for P. aegyptense by Ahmad et al. (2015a, 2015b) from India (see Dos Santos & Avenant-Oldewage (2020) for details), molecular data for both species remain elusive.

Diplozoid species were generally considered to be strictly host specific, with only selected species like Paradiplozoon homoion (Bychowsky & Nagibina, 1959), Paradiplozoon megan (Bychowsky & Nagibina, 1959) and Paradiplozoon bingolense Civáňová, Koyun & Koubková, 2013 infecting several hosts (Khotenovsky, 1985; Matějusová et al., 2002; Benovics et al., 2021; Nejat et al., 2023). However, specificity of diplozoid species appear highly variable, being impacted by limited data, singular reports/collections for several species, taxonomic uncertainty, and unsupported identifications. The infection of more than one host family or order by a diplozoid species is still exceedingly rare, with only a few Diplozoinae species suspected of doing so. It has been noted that diporpa are less selective than adult diplozoids, with Paperna (1963) infecting Oreochromis niloticus (Linnaeus, 1758) with juvenile Paradiplozoon minutum (Paperna, 1964), but they were presumably not able to fuse and mature on incompatible hosts. Paperna (1979) furthermore, collected adult diplozoids from B. macrolepidotus (the type host of P. ghanense) and identified them as P. aegyptense even though this diplozoid was described from a cyprinid host and all other records of this taxon are from cyprinoids. Most collections of diplozoids from non-cyprinoids, including this collection of P. aegyptense from B. macrolepidotus by Paperna (1979), lack detail substantiated by morphological or taxonomic reports.

To improve understanding and differentiate diplozoid taxa in Africa, the morphology of both P. ghanense and P. aegyptense are redescribed here. Some of the specimens studied here likely represent the singular collection of P. aegyptense from B. macrolepidotus by Paperna (1979), enabling re-evaluation of this record. Additionally, a summary of all records for diplozoids occurring on non-cyprinoid hosts is presented, scrutinized and discussed here.

Materials and Methods

Slide mounted material of P. aegyptense and P. ghanense from both the SNMNH and the RMCA were studied using standard compound microscopy techniques, including phase contrast and differential interference contrast (DIC) microscopy. Details for the studied specimens are given in Table 2. The specimens from the RMCA collection were studied at the University of Johannesburg using a Zeiss Axioplan 2 Imaging Light Microscope with Axiovision 4.7.2. The specimens from the SNMNH collection were studied on site in Washington using an Olympus BX51 with Leica Application Suite v4.4.0, or an Olympus BX6F with Olympus CellSens Dimensions 1.13. Morphometric analyses were carried out using either AxioVs40 V 4.8.2.0 or LCmicro 2.4 (Build 29191) software. Measurements were taken following Khotenovsky (1985) and Pugachev et al. (2010). Measurements of structures not in the correct orientation (for example clamps in lateral view), or those that were damaged, were excluded. All measurements are in μm unless otherwise noted, providing three significant figures only. Measurements are presented as a mean with range in parentheses. Illustrations of the haptoral sclerites were produced using both camera lucida illustrations and from digital photomicrographs, using Corel DRAW® Graphics Suite X6 software to digitally create them. Sclerite terminology (Fig. S1) was adapted from Nejat et al. (2023). The obtained data were compared to the descriptions of P. ghanense and P. aegyptense by Thomas (1957) and Fischthal & Kuntz (1963) respectively, as well as the summaries by Khotenovsky (1985). Morphometric data of P. aegyptense were also compared to the data for this species by Ahmad et al. (2015a) and that of P. homoion by Bychowsky & Nagibina (1959), Khotenovsky (1985) and Matějusová et al. (2002). All literature and collection records of Diplozoinae from Africa were summarized and tabulated, as well as all collection records of diplozoids from hosts other than Cypriniformes.

Table 2 Details of the diplozoid specimens studied here from the collections of the Smithsonian National Museum of Natural History (SNMNH), Washington, USA and the Royal Museum of Central Africa (RMCA), Tervuren, Belgium.

	Paradiplozoon ghanense (Thomas, 1957)	Paradiplozoon aegyptense (Fischthal & Kuntz, 1963)	
	USNM 1367213	USNM 1548457	RMCA_VERMES_35512*	USNM 1355449	USNM 1355450	USNM 1363700	RMCA_VERMES_35196	RMCA_VERMES_35580	
Alternate	USNPC # 071636	USNPC # 071636	M.T. 35.512	USNPC # 059653	USNPC # 059654	USNPC # 068090	M.T. 35.196	M.T. 35.580	
Collection name	SNMNH	SNMNH	RMCA	SNMNH	SNMNH	SNMNH	RMCA	RMCA	
Previous collection	US National Parasite Collection	US National Parasite Collection	–	US National Parasite Collection	US National Parasite Collection	US National Parasite Collection	–	–	
Recorded as	Diplozoon ghanense Thomas, 1957	Diplozoon ghanense Thomas, 1957	Diplozoon aegyptensis Fischthal & Kuntz, 1963	Diplozoon aegyptensis Fischthal & Kuntz, 1963	Diplozoon aegyptensis Fischthal & Kuntz, 1963	Diplozoon aegyptensis Fischthal & Kuntz, 1963	Diplozoon aegyptensis Fischthal & Kuntz, 1963	Diplozoon aegyptensis Fischthal & Kuntz, 1963	
Status	Lectotype**	Paralectotype**	Voucher	Holotype***	Paratypes	Paratypes	Paratype****	Voucher	
Specimen	1 Adult	1 Adult	1 Adult + 1 Diporpa	1 Adult	2 Adults	5 Adults	1 Adult	1 Adult (Damaged)	
Collector	J.D. Thomas	J.D. Thomas	I. Paperna	R.E. Kuntz	R.E. Kuntz	R.E. Kuntz	J.H. Fischthal	I. Paperna	
Identified by	J.H. Fischthal	J.H. Fischthal	–	J.H. Fischthal & R.E. Kuntz	J.H. Fischthal & R.E. Kuntz	J.H. Fischthal & R.E. Kuntz	–	–	
Date collected	Apr 1956	Apr 1956	1973	6 Sep 1952	6 Sep 1952	Aug 1953	1952	–	
Date identified	1970	1970	1973	8 Feb 1962	8 Feb 1962	Jan 1963	1959	1969	
Location	Black Volta River, Lawra Ghana	Black Volta River, Lawra Ghana	Lake Albert, Butiaba, Uganda	Giza Fish Market, Cairo, Egypt	Giza Fish Market, Cairo, Egypt	Giza Fish Market, Cairo, Egypt	Giza Fish Market, Cairo, Egypt	Lake Volta, Black and White Volta Confluence, Ghana	
GPS	–	–	N 01° 8′ E 31° 32′	N 30° 05′ E 31° 12′	N 30° 05′ E 31° 12′	N 30° 05′ E 31° 12′	N 30° 05′ E 31° 12′	–	
Site	–	–	Gills	–	–	–	Gills	Gills	
Host family	Alestidae	Alestidae	Alestidae	Cyprinidae	Cyprinidae	Cyprinidae	Cyprinidae	Cyprinidae	
Host	Brycinus macrolepidotus Valenciennes, 1850	Brycinus macrolepidotus Valenciennes, 1850	Brycinus macrolepidotus Valenciennes, 1850	Labeo forskalii Rüppell, 1835	Labeo forskalii Rüppell, 1835	Labeo forskalii Rüppell, 1835	Labeo forskalii Rüppell, 1835	Labeo coubie Rüppell, 1832	
Recorded as	Alestes macrolepidotus
(Valenciennes, 1850)	Alestes macrolepidotus
(Valenciennes, 1850)	–	–	–	–	–	Labeo cubie	
Record	Thomas (1957)	Thomas (1957)	Paperna (1979)	Fischthal & Kuntz (1963)	Fischthal & Kuntz (1963)	Fischthal & Kuntz (1963)	Fischthal & Kuntz (1963)	Paperna (1969)	
Remarks	Corrosive acetate, Ehlrlich’s acid heamotoxylin, Mayer’s carmalum*****	Corrosive acetate, Ehlrlich’s acid heamotoxylin, Mayer’s carmalum*****	–	AFA, Haematoxylin/paracarmine, Canada balsam	AFA, Haematoxylin/carmine, Canada balsam	AFA, Haematoxylin/paracarmine, Canada balsam*****	AFA, Haematoxylin/paracarmine, Canada balsam*****	–	
Notes:

* Originally identified as P. aegyptense but re-identified as P. ghanense here.

** Originally catalogued as vouchers but are designated as lectotype and paralectotype here.

*** Designated as the “type” in the original description.

**** Originally catalogued as a voucher but designated as paratype here.

***** Information from literature and not museum catalogue or slide label.

Results

Both voucher specimens of P. ghanense from the SNMNH had highly similar morphometric and haptoral sclerite details, supporting their conspecificity. However, one of the specimens in lot USNM 1367213 bears a striking resemblance to the illustration by Thomas (1957) and thus the specimens in this lot are likely the specimens used to describe the taxon. This is supported by the fact that the collection dates and localities of the specimens correspond to the description. The specimens were also deposited by J.D. Thomas, even though no designation of types was made in Thomas (1957). As such, we consider the specimens in lot USNM 1367213 as the syntypes of P. ghanense following Articles 72.1.1 and 72.4.1.1 and Recommendation 73F of The International Code of Zoological Nomenclature (ICZN, 1999). Due to the similarity of the illustration in Thomas (1957) to one of the specimens in USNM 1367213, this specimen is designated here as the lectotype for the taxon (Article 74.1, Recommendation 73F and 74B (ICZN, 1999)), with the second specimen becoming the paralectotype (Article 73.2.2, 74.1.3 and 74.4 (ICZN, 1999)), separated into a new lot - USNM 1548457.

Of the 12 specimens designated as P. aegyptense (11 adults and one diporpa), most had similar morphometrics and haptoral sclerite characteristics. However, the hook sizes, clamp sclerites and general morphology of one slide (RMCA_VERMES_35512; one adult and one diporpa) resembled the details for P. ghanense instead. This specimen is thus re-designated as a voucher of P. ghanense, leaving 10 P. aegyptense specimens. USNM 1355449 was designated as the “type” of P. aegyptense in the original description and is catalogued in the SNMNH as such. However, paratypes were also designated by Fischthal & Kuntz (1963), and thus USNM 1355449 is actually the holotype (Article 73.1.1 (ICZN, 1999)). Furthermore, only USNM 1355450 is designated as paratype material in the original description, but USNM 1363700 is designated as paratype material in the SNMNH catalogue, while RMCA_VERMES_35196 shares the same collection and collector details as the type material. Additionally, even though only three specimens (one “type” and two paratypes) are catalogued by Fischthal & Kuntz (1963), the authors state that the “species is represented in the collection by nine pairs of adults in permanent copula” (sic.) suggesting that USNM 1363700 and RMCA_VERMES_35196 are the remaining six specimens of the type series. As such, all material deposited by the species authors are considered part of the type series for P. aegyptense, supporting the designation of USNM 1363700 as paratypes and indicting that RMCA_VERMES_35196 should be designated as a paratype as well. RMCA_VERMES_35580 is now the only voucher specimen of P. aegyptense.

The morphometric data presented here for both P. ghanense and P. aegyptense are based only on the type material of each species, while the voucher material is discussed in the remarks section and their data presented in Tables 3 and 4 respectively. The voucher diporpa of P. ghanense is discussed separately in the morphometric section of this species.

Table 3 Summary of the morphometric data for Paradiplozoon ghanense (Thomas, 1957) based on type series and voucher material studied here, alongside data from the original description and Khotenovsky (1985).

	Lectotype
n = 1	Type series
n = 2	Voucher
n = 1	Thomas (1957)	Khotenovsky (1985) *	Diporpa
n = 1	
Total body length	3,410 (3,240–3,580)	3,777 (3,240–4,250)	2,350 (2,170–2,540)	3,210–3,830	3,200–3,800	1,000	
Anterior length	1,970 (1,850–2,100)	2,300 (1,850–2,810)	1,210 (1,100–1,320)	1,860–2,540	1,900–2,500	537	
Anterior width	656 (636–677)	694 (636–754)	431 (411–451)	640–730	–	288	
Posterior length	657 (624–691)	664 (596–755)	611 (560–667)	380–480	400–500	464	
Posterior width	119 (92.9–151)	136 (93–226)	170 (140–208)	380–480	–	201	
Clamps length	97 (80.4–106)	95.8 (80.4–106)	65.6 (59.5–75)	100–110**	100–110	45.2 (39.7–51.4)	
Clamps width	150 (132–162)	140 (105–162)	94.5 (84.2–107)	120–160**	120–160	66.5 (61.6–73.8)	
Hook length	21.2 (21–21.3)	21.5 (21–21.9)	21.7 (21.2–22.1)	–	–	20.8 (20.6–21.1)	
Handle	39.2 (39.1–39.4)	40.3 (39.1–42.6)	40.7 (38.6–42)	–	–	41.7	
Haptor length	376 (325–436)	467 (325–661)	533 (497–571)	420–650	–	272	
Haptor width	585 (569–602)	494 (362–602)	336 (332–340)	270–620	–	285	
Fusion region length	625	563 (508–625)	428.4	380–480	–	–	
Fusion region width	925	951 (925–977)	639.4	900–920	–	–	
Space between suckers	18.4 (17.2–19.7)	21.4 (17.2–29.3)	10.9 (5.1–23.7)	–	–	11.5	
Sucker length	59 (56.3–62.5)	62.5 (56.3–68.5)	48.6 (45.3–52)	50–75	–	55.2 (54.5–56.1)	
Sucker width	55.8 (53–58.8)	57.6 (52.9–67.9)	55.8 (54.2–57.4)	50–70	–	53.6 (53.4–53.8)	
Sucker to anterior	20.5 (14.6–28.8)	24.9 (14.6–32.8)	20.6 (16.2–26.3)	30–35	–	28.4	
Pharynx length	55.1 (48.3–62.9)	54.7 (45.4–65.2)	57.2 (54.2–60.5)	45–70	45–70	59.9	
Pharynx width	41.1 (38–44.6)	43.1 (38–45.4)	46.7 (45.8–47.7)	40–50	40–50	48.2	
Prepharynx length	17.1 (15.8–18.5)	22.9 (15.8–31.8)	29.1 (25.6–33.1)	5–25	–	20.6	
Egg length	258	258	–	260	260	–	
Egg width	114	114	–	115	115	–	
Anoperculum	167	167	–	–	–	–	
Testes length	152 (152–152)	134 (123–152)	84.4 (82.1–86.8)	120–160	–	–	
Testes width	123 (123–123)	118 (115–123)	76.3 (70.3–82.9)	110–120	–	–	
Ovary length	193 (193–193)	281 (193–395)		360–400	–	–	
Ovary width	327 (327–327)	233 (178–327)		170–240	–	–	
Clamp 1 length	92.7 (90.5–95.1)	94.4 (90.5–102)	67.1 (63.3–71)	–	–	51.3 (51.2–51.4)	
Clamp 1 width	133 (132–134)	125 (105–138)	89.2 (84.2–94.6)	–	–	64.2 (63.5–65)	
Clamp 2 length	101 (97.7–104)	100 (93.3–104)	67.5 (61.2–74.9)	–	–	45.6 (43.6–47.7)	
Clamp 2 width	155 (143–162)	144 (123–162)	101 (94.9–107)	–	–	71.8 (69.8–73.8)	
Clamp 3 length	102 (93.2–106)	97.7 (87.6–106)	65.1 (59.9–68.9)	–	–	42.4 (42.3–42.5)	
Clamp 3 width	156 (152–158)	147 (126–158)	98.1 (91.6–105)	–	–	67.6 (65.7–69.6)	
Clamp 4 length	91.4 (80.4–98)	92.2 (80.4–98)	62.7 (59.5–66.7)	–	–	42.2 (39.7–44.8)	
Clamp 4 width	150 (142–162)	142 (122–162)	90.2 (88.5–94.3)	–	–	62.8 (61.6–64.1)	
Notes:

* Summary of description.

** Values switched to match convention. See P. ghanense remarks section.

Table 4 Summary of the morphometric data for Paradiplozoon aegyptense (Fischthal & Kuntz, 1963).

Morphometric data for the type series and voucher material studied here, alongside data from the original description; the summary by Khotenovsky (1985); data for P. aegyptense by Ahmad et al. (2015a); and data for Paradiplozoon homoion (Bychowsky & Nagibina, 1959) by Bychowsky & Nagibina (1959), Khotenovsky (1985) and Matějusová et al. (2002).

	Paradiplozoon aegyptense
(Fischthal & Kuntz, 1963)	Paradiplozoon homoion
(Bychowsky & Nagibina, 1959)	
	Holotype
n = 1	Type series
n = 9	Voucher
n = 1	Fischthal & Kuntz (1963)	Khotenovsky (1985) *	Ahmad et al. (2015a)	Bychowsky & Nagibina (1959)	Khotenovsky (1985)	Matějusová et al. (2002)	
Total body length	5,060 (4,840–5,300)	4,370 (3,640–6,150)	4,470 (4,350–4,590)	4,530 (3,620–5,770)	3,600–5,800	4,200 (3,950–4,250)	3,000–5,300	3,100 (1,800–5,200)	–	
Anterior length	3,340 (3,230–3,450)	2,630 (1,920–3,830)	2,520 (2,510–2,540)	2,670 (1,880–3,450)	1,900–3,500	–	–	1,800 (800–3,000)	–	
Anterior width	796 (769–823)	572 (305–925)	903 (821–993)	558 (299–836)	–	–	700–1,400*	–	–	
Posterior length	1,410 (1,240–1,610)	1,080 (839–1,840)	1,080 (972–1,200)	1,130 (867–1,870)	90–1,900	–		1,000 (500–2,000)	–	
Posterior width	225 (198–257)	183 (120–257)	184 (177–191)	178 (130–245)	–	–	700–1,400*	–	–	
Clamps length	74.2 (69.4–78.3)	72.5 (55.7–85.4)	51.5 (43.7–63.1)	70 (65–79)	65–79	114 (100–124)	–	–	–	
Clamps width	93.4 (80–106)	97.9 (73.2–120)	76.9 (64.4–82.5)	97 (92–102)	92–102	45 (40–48)	–	–	–	
Hook length	18.9 (18.9–19)	18.6 (17.6–19.7)	18.2	16.5 (16–17)	16–17	–	17	18–22	19–22.4	
Handle	46.2 (45.5–47)	44.9 (41.1–47.9)	–	49 (48–49)	48–49	–	42	38–47	–	
Haptor length	331 (325–336)	330 (281–412)	345 (326–365)	298 (253–360)	–	–	–	–	–	
Haptor width	330 (276–394)	270 (189–399)	272 (269–274)	–	–	–	–	–	–	
Fusion Region length	442	406 (320–459)	461	376 (291–437)	–	–	–	–	–	
Fusion Region width	1,090	876 (733–1,098)	1,150	727 (652–874)	–	–	–	–	–	
Space between suckers	–	8.93 (4.45–12.6)	9.57 (6.21–14.8)		–	–	–	–	–	
Sucker length	120 (116–126)	106 (82.9–128)	73.8 (67.9–79)	110 (95–125)	95–125	48 (32–64)	70–90	70 (49–108)	–	
Sucker width	89.9 (87.3–93.7)	95.1 (75.5–122)	78.8 (76.7–80)	95 (78–103)	78–103	48 (32–64)	60–80	65 (49–81)	–	
Sucker to anterior	46.6 (44.4–49)	37.5 (27.1–49)	21.2 (21.1–21.2)	38 (29–46)	–	–	–	–	–	
Pharynx length	71.2 (71.2–71.2)	63.2 (51.8–83.2)	61 (51.2–72.8)	62 (51–75)	51–75	64 (56–72)	70–80	70 (49–81)	–	
Pharynx width	46.5 (45.4–47.6)	43.3 (39.1–50.3)	51.4 (46.8–56.5)	44 (40–50)	40–50	47 (44–50)	70–80	49 (33–65)	–	
Prepharynx length	23.6 (22.2–25.1)	25.2 (13.8–32.8)	15.9 (15.9–16)	27 (20–34)	–	54 (41–68)	–	–	–	
Egg length	298 (298–298)	280 (239–308)	305	292 (254–313)	158–187	250 (220–280)	250–280	260–300	–	
Egg width	127 (122–132)	104 (84.6–132)	114	107 (81–132)	81–132	82 (76–88)	100–150	81–124	–	
Anoperculum	167 (167–167)	170 (149–191)	183	170 (158–187)**	–	–	–	–	–	
Testes length	121 (104–141)	127 (100–187)	125	136 (103–190)	–	155 (140–170)	–	–	–	
Testes width	81.1 (73.7–89.1)	81.8 (63.4–94.7)	107	80 (63–93)	–	105 (100–110)	–	–	–	
Ovary length	390 (342–444)	339 (257–470)	423	359 (276–460)	–	–	–	–	–	
Ovary width	179 (145–221)	166 (121–225)	297	183 (103–202)	–	–	–	–	–	
Clamp 1 length	72.3 (71.8–73.5)	68.9 (61.6–84.8)	45.9 (43.7–48.5)	–	–	–	–	65 (54–87)	31–101	
Clamp 1 width	90 (80–96.3)	88.7 (73.2–104)	69.6 (64.4–75.8)	–	–	–	120–140	103 (87–162)	70–143	
Clamp 2 length	75.1 (69.4–78.3)	73.6 (63.2–81.1)	52 (50.6–54.1)	–	–	–	–	76 (54–87)	66–96	
Clamp 2 width	95.5 (83.7–106)	101 (82.4–116)	80.1 (76.6–82.5)	–	–	–	150–180	141 (114–184)	100–180	
Clamp 3 length	73.9 (71.9–76)	74.1 (58.3–82.5)	53.7 (50.3–57)	–	–	–	–	76 (54–92)	58–99	
Clamp 3 width	96.3 (94.3–98.3)	103 (88–120)	80.8 (79.5–81.8)	–	–	–	160–200	146 (114–200)	109–199	
Clamp 4 length	76.9 (76.2–77.6)	73.7 (55.7–85.4)	54.7 (48.9–63.1)	–	–	–	–	81 (65–92)	56–102	
Clamp 4 width	92.3	101 (87.1–117)	77.7 (77.2–78.3)	–	–	–	160–200	146 (114–206)	85–197	
Notes:

* Duplication of description.

** Measurement for operculum in Fischthal & Kuntz (1963). See P. aegyptense remarks section.

Family: Diplozoidae Palombi, 1949

Subfamily: Diplozoinae Palombi, 1949

Genus: Paradiplozoon Akhmerov, 1974

Paradiplozoon ghanense (Thomas, 1957)

Synonym: Diplozoon ghanense Thomas, 1957

Type host: Brycinus macrolepidotus Valenciennes, 1850

Other hosts: Alestes baremoze (Joannis, 1835)

Type locality: Black Volta, Lawra, Ghana

Other localities: Lake Volta, Ghana; Lake Albert, Uganda; Anambra River, Nigeria (See Table 1 for detail).

Infection site: Gills

Lectotype: USNM 1367213 (one slide, USNPC 071636)

Paralectotype: USNM 1548457 (one slide, USNPC 071636)

Voucher: RMCA_VERMES_35512 (one slide, M.T. 35.512)

Morphology

Adult (n = 2; Figs. 1, S2 and S3; Tables 3 and S1): Specimens permanently fused in cross-copula. Body 3,780 (3,240–4,250). Broad, dorsoventrally flattened anterior, 2,300 (1,850–2,810) long, 694 (636–754) wide, tapered toward oral and fusion areas. Sub-cylindrical fusion area, 563 (508–625) long, 951 (925–977) wide. Posterior subcylindrical, 664 (596–755) long, 136 (93–226) wide. Widened haptor at posterior end, 467 (325–661) long, 494 (362–602) wide, without any conspicuous features or protrusions. Clear constriction between posterior and fusion area. Tegument with small, delicate plicae, slightly larger in posterior between fusion area and haptor, not pronounced. No dilations in posterior nor any diagnostically pronounced plicae. Oral opening “U”-shaped, sub-terminal, ventral. Buccal cavity with two suckers, 62.5 (56.3–68.5) long, 57.6 (52.9–67.9) wide, close to each other, 21.4 (17.2–29.3) between them. Suckers 24.9 (14.6–32.8) from anterior. Prepharynx short, 22.9 (15.8–31.8). Pharynx, 54.7 (45.4–65.2) long, 43.1 (38–45.4) wide, posterior to suckers. Intestine single caecum from pharynx to posterior, approaching haptor, diverticula in anterior only. Vitellaria densely packed from just posterior of pharynx to mostly anterior of fusion area.

Figure 1 Illustrations of the egg and haptoral sclerites of Paradiplozoon ghanense (Thomas, 1957) from Brycinus macrolepidotus Valenciennes, 1850.

(A) Egg of lectotype USNM 1367213. (B) Third clamp of paralectotype USNM 1548457 with isolated (C) anterior clamp jaw, (D) posterior clamp jaw, (E) anterior of median sclerite, and (F) posterior of median sclerite. (G) Central hook of paratype USNM 1548457.

Gonads entirely in fusion area, single testes of one specimen close to anterior body. Ovary anterior to testes, long, folded, 281 (193–395) long, 233 (178–327) wide. Testes post-ovarian (but in contact or slightly overlapping ovary), conspicuous, round to oval, single, smooth, 134 (123–152) long, 118 (115–123) wide. Egg large, spindle shaped (Fig. 1A), 258 long, 114 wide. Long, coiled filament attached to operculum, anoperculum 167 long.

Haptor with two rows of ventrally directed clamps, four in a row on opposite ventrolateral margins. Clamps 95.8 (80.4–106) long, 140 (105–162) wide on average. First clamp 94.4 (90.5–102) long, 125 (105–138) wide, second clamp 100 (93.3–104) long, 144 (123–162) wide, third clamp 97.7 (87.6–106) long, 147 (126–158) wide, fourth clamp 92.2 (80.4–98) long, 142 (122–162) wide. Clamps consist of a pair of “J”-shaped anterior clamp jaws, a pair of posterior clamp jaws and a ‘‘U’’-shaped median plate (Fig. 1B). Clamp jaws (Figs. 1C and 1D) slender but not delicate. Lateral sclerite absent. Anterior of median sclerite rounded, with elongate obovate to spoon-shaped anterior spur extending towards anterior clamp jaws (Fig. 1E). No anterior joining sclerite between anterior clamp jaws and anterior end of median sclerite. Board trapezoid dorsal joining sclerite above anterior spur (Fig. 1B and S3A, S3B). Tendon guiding termination at posterior end of median sclerite claw-shaped with large central opening (Fig. 1F). Single posterior joining sclerite with broad base and rounded anterior sometimes visible between posterior clamp jaws and posterior of median sclerite.

Central hooks present (Fig. 1G), single pair, each roughly at level of first clamp towards medial aspect of haptor. Hook 21.5 (21–21.9), handle 40.3 (39.1–42.6). Wing mostly overlaying blade of hook.

Diporpa (n = 1; Figs. S4A, S4B and S5A; Tables 3 and S1): Body length 1,000. Broad, dorsoventrally flattened anterior, 537 long, 288 wide, tapered towards anterior. Fusion area absent. Slight constriction between anterior and posterior. Posterior subcylindrical, 464 long, 201 wide, no pronounced dilations. Widened haptor at posterior, 271 long, 285 wide, without any conspicuous features or protrusions. Tegument with small, fine plicae, slightly more noticeable in anterior of posterior, not pronounced. Oral opening “U”-shaped, sub-terminal, ventral. Buccal cavity with two suckers, 55 (55–56) long, 55 (52–59) wide, close to each other, 11.5 between them. Suckers 28 from anterior. Suckers 28.4 from anterior. Prepharynx short, 20.6 long. Pharynx, 59.9 long, 48.2 wide, posterior to suckers. Intestine single caecum from pharynx to middle of haptor. Vitellaria and gonads absent, only primordial mass present.

Haptor with two rows of ventrally directed clamps, four in a row on opposite ventrolateral margin. Clamps 45.2 (39.7–51.4) long, 66.5 (61.6–73.8) wide on average. First clamp 51.3 (51.2–51.4) long, 64.2 (63.5–65) wide, second clamp 45.6 (43.6–47.7) long, 71.8 (69.8–73.8) wide, third clamp 42.4 (42.3–42.5) long, 67.6 (65.7–69.6) wide, fourth clamp 42.2 (39.7–44.8) long, 62.8 (61.6–64.1) wide. Clamp sclerites highly similar to type material (Figs. S4A and S4B). However, anterior clamp jaws extend anteriorly at their medial connection, forming broad connection dorsal to anterior protrusion of medial sclerite.

Central hooks present, single pair, each roughly at level or anterior to first clamp towards medial aspect of haptor. Hook 20.8 (20.6–21.1), handles 41.7.

Remarks: The morphometric data obtained from the newly identified type series of P. ghanense were similar to that in the original description (Table 3), with only slightly wider ranges recorded for some features. However, even though specimen USNM 1367213 (now only one specimen and designated as lectotype) bore a striking resemblance to the illustration in the description, damage at the extremity of one haptor and the pre-haptoral region of the other were observed (Fig. S2), which was not captured in the illustration by Thomas (1957). Similarly, some of the clamps on both haptors were damaged or missing, with the illustration by Thomas (1957) displaying all clamps intact. Whether damage to the specimen occurred after the description, possibly due to remounting, or whether this damage was edited out during illustration is not known. Unfortunately, the clamps and hooks of the lectotype were not ideally orientated for illustration, but the same morphology was observed in both type series specimens. As such, the third clamps and hook of the paralectotype are illustrated in Fig. 1. The hooks of the taxon could be observed and are described here for the first time. Thomas (1957) does not state how many parasites were collected or studied from the two infected fish, thus the two specimens studied here may be the only material in the type series.

Specimen RMCA_VERMES_35512, which is designated as a voucher of P. ghanense here, is morphologically nearly identical to that of the type series material (Figs. S5B–S5D). This includes the details of the clamp sclerites and the size of the hooks (Figs. S4C–S4E), supporting the re-designation of this material as P. ghanense. However, large deviations in the sizes of the body (total, anterior length and width), fusion area size, gonad size, and especially the sizes of the clamps were observed. Nevertheless, the similarity of the more taxonomically informative clamp sclerites and hook sizes validate the re-identification of the material. The hooks of the diporpa could not be studied in an ideal orientation and thus only their size could be recorded.

Using the keys for Paradiplozoon taxa in South-East Asia and Africa by Khotenovsky (1985), the supplemented data for P. ghanense does not match any of the criteria as there is a gap where taxa with hooks between 20 and 25 μm should fall. However, if the assumption is made that this gap is unintentional and that the 11th line should refer to taxa with hooks larger than 20 μm instead of larger than 25 μm, then the specimens studied here conform to the identification as P. ghanense. The clamp morphology of P. ghanense differs from all other diplozoid taxa. The most striking difference is the absence of a lateral sclerite. The division of the posterior clamp jaw to form a lateral sclerite is present in most diplozoid taxa, except for P. bingolense from Turkey and Paradiplozoon iraqense Al-Nasiri & Balbuena, 2016 from Iraq. The gross clamp morphology of both P. bingolense and P. iraqense are similar to that of P. ghanense but are more robust. Among African Paradiplozoon, the clamp sclerites of P. ghanense are most similar to those of P. moroccoense, but the latter species has lateral sclerites. Paradiplozoon ghanense differs from all species, as well as all other African Paradiplozoon spp., in terms of general body shape, all lacking the clear constriction between the fusion area and the posterior, as well as restriction of the gonads to the fusion region. Most also have smaller hooks than P. ghanense. The confinement of the reproductive organs to the fusion area in P. ghanense is only shared by Sindiplozoon Khotenovsky, 1981 species.

Paradiplozoon aegyptense (Fischthal & Kuntz, 1963)

Synonym: Diplozoon aegyptensis Fischthal & Kuntz, 1963

Type host: Labeo forskalii Rüppell, 1835

Other hosts: Enteromius paludinosus (Peters, 1852), Labeo coubie Rüppell, 1832, Labeo cylindricus Peters, 1852, Labeo victorianus Boulenger, 1901, Raiamas senegalensis (Steindachner, 1870), Labeo sp.

Unverified hosts: Cyprinus carpio Linnaeus, 1758, Schizopyge niger (Heckel, 1838), Schizothorax progastus (McClelland, 1839), Carassius carassius (Linnaeus, 1758)

Type locality: Giza Fish Market, Cairo, Egypt

Other localities: Lake Volta, Ghana; Lake Albert and Aswa River, Uganda; Nzoia River, Kenya; Ruaha River, Tanzania.

Unverified localities: Several localities in India (See Table 1 for detail).

Infection site: Gills

Holotype: USNM 1355449 (one slide, USNPC 059653)

Paratypes: USNM 1355450 (two slides, USNPC 059654); USNM 1363700 (five slides, USNPC 068090); RMCA_VERMES_35196 (one slide, M.T. 35.196)

Voucher: RMCA_VERMES_35580 (one slide, M.T. 35.580)

Morphology

Adult (n = 9; Fig. 2; Tables 4 and S2): All specimens permanently fused in cross-copula. Body 4,370 (3,640–6,150) long. Broad, dorsoventrally flattened anterior, 2,630 (1,920–3,830) long, 572 (305–925) wide, tapered toward oral and fusion areas. Sub-cylindrical fusion area, 406 (320–459) long, 876 (733–1098) wide. Posterior subcylindrical, 1,080 (839–1840) long, 183 (120–257) wide. Simple, disk-like haptor at posterior, 330 (281–412) long, 270 (189–399) wide, without conspicuous features or protrusions. No constriction between posterior and fusion area, no dilation of posterior. Tegument with small, delicate plicae, slightly more conspicuous in posterior between fusion area and haptor, not pronounced. Oral opening “U”-shaped, sub-terminal, ventral. Buccal cavity with two suckers, 106 (82.9–128) long, 95.1 (75.5–122) wide, close to each other, often touching, 8.93 (4.45–12.6) apart on occasion. Suckers 37.5 (27.1–49) from anterior margin. Prepharynx short, 25.2 (13.8–32.8). Pharynx, 63.2 (51.8–83.2) long, 43.3 (39.1–50.3) wide, posterior to suckers. Intestine single caecum from pharynx to anterior, occasionally reaching slightly into haptor, diverticula in anterior but not in posterior. Vitellaria densely packed from just posterior to pharynx to anterior of gonads in fusion area.

Figure 2 Illustrations of the egg and haptoral sclerites of Paradiplozoon aegyptense (Fischthal & Kuntz, 1963) holotype USNM 1355449 from Labeo forskalii Rüppell, 1835.

(A) Egg. (B) Third clamp with isolated (C) anterior clamp jaw, (D) posterior clamp jaw, (E) anterior of median sclerite, (F) variations of anterior joining sclerites, and (G) posterior of median sclerite. (H) Central hook.

Gonads mainly in fusion area, testes partly or entirely in posterior. Ovary anterior to testis, long, folded, 339 (257–470) long, 166 (121–225) wide. Testes post-ovarian (but in contact or slightly overlapping ovary), conspicuous, round to oval, single, smooth, 127 (100–187) long, 81.8 (63.4–94.7) wide. Eggs large, spindle shaped (Fig. 2A), 280 (239–308) long, 104 (84.6–132) wide. Long, coiled filament attached to operculum, anoperculum 170 (149–191).

Haptor with two rows of ventrally directed clamps, four in a row on opposite ventrolateral margins. Clamps 72.5 (55.7–85.4) long, 97.9 (73.2–120) wide on average. First clamp 68.9 (61.6–84.8) long, 88.7 (73.2–104) wide, second clamp 73.6 (63.2–81.1) long, 101 (82.4–116) wide, third clamp 74.1 (58.3–82.5) long, 103 (88–120) wide, fourth clamp 73.7 (55.7–85.4) long, 101 (87.1–117) wide. Clamps consist of a pair of “J”-shaped anterior clamp jaws, a pair of posterior clamp jaws and a ‘‘U’’-shaped median sclerite (Fig. 2B). Clamp jaws thick and robust (Figs. 2C and 2D), lateral sclerite usually visible. Median sclerite rounded anteriorly with rectangular to truncate anterior spur (Fig. 2E). Trapezoid to rounded anterior joining sclerite between anterior projection of median sclerite and anterior clamps jaws, with a pair of small rectangular anterior distal joining sclerite sometimes visible between anterior joining sclerite and anterior clamp jaws (Fig. 2F). Board trapezoid dorsal joining sclerite observed above anterior of median sclerite (Figs. 2B and S3C–S3F). Tendon guiding termination at posterior end of median sclerite claw-shaped, with large central opening (Fig. 2G). Single, narrow posterior joining sclerite between anterior clamp jaw and posterior end of median sclerites, wider at base with rounded anterior end.

Central hooks present (Fig. 2H), single pair, roughly at level of first clamp towards medial aspect of haptor, ventral. Hook 18.6 (17.6–19.7), handle 44.9 (41.1–47.9). Wing sometimes extending over blade of hook.

Remarks: The morphometric data generated here based on the type series were highly similar to that in the original description of P. aegyptense (Table 4), with only slightly wider or narrower ranges recorded for some features. The only notable difference between the measurements generated here and the original description was the larger size of the hooks, 16.5 (16–17) in the original description and 18.7 (17.7–19.1) here. The similarity in the data generated for all material considered part of the type series here support the designation of USNM 1363700 and RMCA_VERMES_35196 as paratype material even though this material is not specifically designated as such in the original description. Thus, the entire type series of nine adults indicated by Fischthal & Kuntz (1963) was studied here. Some of the clamps on one haptor of the holotype were not fully formed, or malformed, unlike the illustration in the description (and Khotenovsky (1985)), suggesting these anomalies were edited out of the illustrations.

Voucher specimen RMCA_VERMES_35580 was morphologically highly similar to the data for the type series. The only differences were smaller or larger ranges for some structures (e.g., clamps and suckers), but ranges still overlapped with the type series. Some of the internal organs could not be clearly observed as the voucher material was overstained. Unfortunately, the mounting medium of the voucher had badly deteriorated, necessitating remounting. The cracked medium also caused the specimen to fragment, but all pieces of the specimen were retained, remounted, and imaged fragments digitally recombined (Fig. S6). Destaining was attempted before remounting, but to no avail.

Using the keys for Paradiplozoon taxa in South-East Asia and Africa by Khotenovsky (1985), all specimens studied here conform to the identification as P. aegyptense. The morphology of P. aegyptense differs from other African Paradiplozoon spp. mainly based on the clamp sclerite detail, with P. krugerense most similar. However, P. aegyptense has a single posterior joining sclerite while P. krugerense has a proximal and distal posterior joining sclerite. Additionally, P. krugerense does not display the dorsal joining sclerite or the small rectangular distal joining sclerite sometimes visible between the anterior joining sclerite and anterior clamp jaws of P. aegyptense. The lateral sclerite also appears much smaller in relation to the posterior clamp jaws in P. aegyptense than in P. krugerense. Interestingly, a posterior clamp jaw of one of the third clamps of the holotype was not divided to display a lateral sclerite (Figs. 2B and 2D), a characteristic shared by very few diplozoid taxa (see P. ghanense remarks section). This appears to be an anomaly as it was not seen in the other clamps of the holotype or paratypes, indicating that the clamp did not develop normally and suggesting that the clamp sclerites of the holotype may not be the ideal representative for the taxon. The hook length of P. krugerense (18 (17–19)), and to a lesser extent P. vaalense (19 (18–20)), is similar to the updated data for P. aegyptense (18.7 (17.7–19.1)) presented here. Outside of Africa, the clamp sclerites of P. aegyptense are dissimilar to all diplozoid taxa. This includes P. homoion, in which the trapezoid anterior spur of the median sclerite attaches to the anterior clamp jaws. The testes of P. homoion are also lobed, unlike the smooth testes of P. aegyptense. The hook sizes of P. homoion (19–22.4 in Matějusová et al. (2002)) are generally larger than the data for P. aegyptense, supporting the distinctness of the two taxa. However, the hooks were recorded as 17 in the original description of P. homoion, smaller than those of P. aegyptense recorded here, but similar the those in the description of P. aegyptense (16.5 (16–17) in Fischthal & Kuntz (1963)).

Host specificity in Africa

As can be seen from Table 1, P. aegyptense is the only Paradiplozoon in Africa that has been recorded from more than one host order (Cypriniformes and Characiformes), as well as two cyprinoid families, Danionidae and Cyprinidae. The only record of P. aegyptense being collected from an alestid host matches the details for RMCA_VERMES_35512 studied and redesignated as P. ghanense here. As such, P. aegyptense is now considered specific to members of Cypriniformes only, occurring on four Labeo Cuvier, 1816, one unidentified Labeo, one Enteromius Cope, 1867, and the danionid Raiamas senegalensis (Steindachner, 1870) in Africa. Paradiplozoon ghanense is now the only diplozoid recognised to occur on alestid hosts and the only diplozoid in Africa to occur on Characiformes. This species occurs on two previously congeneric hosts, one Alestes Müller & Troschel, 1844 and one Brycinus Valenciennes, 1850. The unidentified Diplozoon sp. collected by Thurston (1970) was also from Alestes sp., but the authors note its resemblance to P. ghanense. All other Paradiplozoon in Africa are specific to cyprinids, with three of these species currently known from two congeneric hosts each: P. ichthyoxanthon from two Labeobarbus Rüppell, 1835 species, and P. vaalense and P. krugerense from two species of Labeo each. Paradiplozoon moroccoense has only been collected from a single Luciobarbus Heckel, 1843 species. The unidentified Paradiplozoon sp. reported by Truter et al. (2023) occurs on one member of two cyprinid genera each, one Sedercypris Skelton, Swartz & Vreven, 2018 and one Cheilobarbus Smith, 184 species.

Diplozoids infecting non-cyprinoids

A total of 23 sources were found reporting the collection of diplozoids from non-cyprinoid hosts (Table 5). These sources note the collection of diplozoids from 11 orders (and three suborders) other than Cypriniformes, with hosts from 16 families and 21 genera. The majority of the records were for unidentified Diplozoidae or Diplozoon paradoxum von Nordmann, 1832, with some specifically noting that only diporpa larvae were collected. Records reporting on D. paradoxum (especially those prior to Khotenovsky (1985)) are treated as unidentified diplozoids here as this is the go-to identification for diplozoids and these works do not include support for the identification. Many of these records (either unidentified diplozoids or D. paradoxum) were from Khotenovsky (1985), which appears to be a summary of available literature at the time but without citations to the source material. Literature corresponding to some reports by Khotenovsky (1985) could be found, but not all. Additionally, several of the records summarised here are from unpublished theses and thus not peer reviewed (Paperna, 1963; Al-Niaeemi, 1997; Abdul-Rahman, 1999; Al-Sa’adi, 2007; Al-Janae’e, 2010; Al-Jubori, 2013).

Table 5 Collection records of diplozoids from host other than Cypriniformes.

Order	Family	Species	Diplozoid	Locality	Reference	
Acipenseriformes	Acipenseridae	Acipenser gueldenstaedtii Brandt & Ratzeburg, 1833	Diplozoinae gen. sp.	Volga River	Khotenovsky (1985) *	
Anguilliformes	Anguillidae	Anguilla anguilla (Linnaeus, 1758)	Diplozoon paradoxum von Nordmann, 1832	Vistula Lagoon, Baltic Sea	Zaostrovtseva & Evdokimova (2008), Zaostrovtseva (2009)	
Characiformes	Alestidae	Alestes baremoze (Joannis, 1835)	Paradiplozoon ghanense (Thomas, 1957)	Yeji, Volta Lake, Ghana	Paperna (1969)	
Africa	Khotenovsky (1985) *	
Alestes sp.	Diplozoon sp.
(Resembled P. ghanense)	Butiaba, Lake Albert, Uganda	Thurston (1970)	
Brycinus macrolepidotus Valenciennes, 1850	Afrodiplozoon polycotyleus (Paperna, 1973)	Africa	Khotenovsky (1985) *	
Otuocha, Anambra River, Nigeria	Echi & Ezenwaji (2010)	
Paradiplozoon aegyptense (Fischthal & Kuntz, 1963)**	Butiaba, Lake Albert, Uganda	Paperna (1979)	
Africa	Khotenovsky (1985) *	
Paradiplozoon ghanense (Thomas, 1957)	Lawra, Black Volta Riever, Ghana	Thomas (1957)	
Africa	Khotenovsky (1985) *	
Otuocha, Anambra River, Nigeria	Echi & Ezenwaji (2010)	
Characidae	Ctenobrycon spilurus (Valenciennes, 1850)	Paradiplozoon tetragonopterini (Sterba, 1957)	Erfurter Aquarium, Germany	Sterba (1957), Khotenovsky (1985)*	
Gymnocorymbus ternetzi (Boulenger, 1895)	Paradiplozoon tetragonopterini (Sterba, 1957)	Erfurter Aquarium, Germany	Sterba (1957), Khotenovsky (1985)*	
Cichliformes	Cichlidae	Coptodon rendalli (Boulenger, 1897)	Diplozoidae Palombi, 1949
(Nine pars of clamps, possibly new species, likely Neodiplozoinae)	Goma Lakes, Lusaka, Zambia	Batra (1984)	
Coptodon zillii (Gervais, 1848)	Diplozoon paradoxum von Nordmann, 1832	Lagoon ponds near Yeniyurt town, Dörtyol district, Turkey	Yildirim et al. (2010)	
Oreochromis niloticus (Linnaeus, 1758)	Paradiplozoon minutum (Paperna, 1964)
(diporpa larva)	Laboratory conditions, experimental	Paperna (1963)	
Esociformes	Esocidae	Esox lucius Linnaeus, 1758	Diplozoinae gen. sp.	Danube River; Tisza River; Tsimlyansk Reservoir; Barabinskoye Ozero (Барабинские озера)	Khotenovsky (1985) *	
Gadiformes	Lotidae	Lota lota (Linnaeus, 1758)	Diplozoinae gen. sp.	Lake Peipsi; Waters bodies in Ukraine; Vilyuy Dam; Germany; Poland	Khotenovsky (1985) *	
Diplozoon paradoxum von Nordmann, 1832	United Kingdom	Nicoll (1924) *	
Diplozoon sp.	–	Aioanei (1996) *	
Gobiiformes	Gobiidae	Neogobius melanostomus (Pallas, 1814)	Diplozoinae gen. sp.	Dnieper River Delta	Khotenovsky (1985) *	
	Oxudercidae	Periophthalmus waltoni Koumans, 1941	Diplozoon sp.	Marine	Mhaisen & Al-Maliki (1996)	
Mugiliformes	Mugilidae	Planiliza abu (Heckel, 1843)	Diplozoon paradoxum von Nordmann, 1832	Euphrates River, Al-Musaib city, Iraq	Al-Sa’adi (2007) ***	
Mhaisen, Al-Rubaie & Al-Sa’adi (2015)	
Diplozoon sp.	Qamat Ali Canal, Basrah, Iraq	Abdul-Rahman (1999) ***	
Paradiplozoon bliccae (Reichenbach-Klinke, 1961)	Tigris River, Tikrit city, Iraq	Al-Jubori (2013) ***	
Paradiplozoon kasimii (Rahemo, 1980)	Al-Salihiya canal, Basrah, Iraq	Al-Janae’e (2010) ***	
Planiliza subviridis (Valenciennes, 1836)	Paradiplozoon kasimii (Rahemo, 1980)	Qamat Ali canal, Basrah, Iraq	Abdul-Rahman (1999) ***	
Perciformes/
Cottoidei	Cottidae	Cottus gobio Linnaeus, 1758	Diplozoinae gen. sp.	Ukraine; Germany (East and West)	Khotenovsky (1985) *	
Diplozoon paradoxum von Nordmann, 1832	United Kingdom	Nicoll (1924) *	
Perciformes/
Gasterosteoidei	Gasterosteidae	Gasterosteus aculeatus Linnaeus, 1758	Diplozoinae gen. sp.	England	Khotenovsky (1985) *	
Diplozoon paradoxum von Nordmann, 1832	St. Andrews, United Kingdom (Marine)	Nicoll (1915)	
Perciformes/
Percoidei	Percidae	Gymnocephalus cernua (Linnaeus, 1758)	Diplozoinae gen. sp.	Poland	Khotenovsky (1985) *	
Perca fluviatilis Linnaeus, 1758	Diplozoinae gen. sp.	Ili River	Khotenovsky (1985) *	
	Konchozero (Lake); Lake Peipus (Peipsi and Pihkva); Middle Volga Region; Tisza River; Lake in Syr Darya ; Shalqar Koli (Lake)	Khotenovsky (1985) *	
Diplozoon sp.	–	Aioanei (1996) *	
Diplozoon paradoxum von Nordmann, 1832	Poland	Khotenovsky (1985) *	
Vistula Lagoon, Baltic Sea	Zaostrovtseva & Evdokimova (2008), Zaostrovtseva (2009)	
Pregolya River, Russia	Zaostrovtseva & Evdokimova (2008)	
Reka Prokhladnaya (River), Russia	Zaostrovtseva & Evdokimova (2008)	
Paradiplozoon bliccae (Reichenbach-Klinke, 1961)
(As Diplozoon gussevi Gläser, 1964)	Mušov Reservoir (Vodní Nádrž Nové Mlýny-horní), Czech Republic	Lucký, Navrátil & Jirásková (1989)	
Sander lucioperca (Linnaeus, 1758)	Diplozoinae gen. sp.	Dnieper River	Khotenovsky (1985) *	
Sander volgensis (Gmelin, 1789)	Diplozoinae gen. sp.	Don River	Khotenovsky (1985) *	
Siluriformes	Siluridae	Silurus glanis Linnaeus, 1758	Diplozoon sp.
(diporpa larva)	Tigris River, Mosul city, Iraq	Al-Niaeemi (1997) ***	
Paradiplozoon pavlovskii (Bychowsky & Nagibina, 1959)	Tigris River, Mosul city, Iraq	Al-Niaeemi (1997) ***	
Heteropneustidae	Heteropneustes fossilis (Bloch, 1794)	Diplozoon sp.	Qamat Ali Canal, Basrah, Iraq	Abdul-Rahman (1999) ***	
Synbranchiformes	Mastacembelidae	Mastacembelus mastacembelus (Banks & Solander, 1794)	Diplozoon sp.	Qamat Ali Canal, Basrah, Iraq	Abdul-Rahman (1999) ***	
Notes:

* Record not specific, only part of list or review.

** Paradiplozoon ghanense based on the current findings.

*** Unpublished thesis.

The collection of P. ghanense and Afrodiplozoon polycotyleus (Paperna, 1973) from characoids in Africa matches the available knowledge for these taxa, with the collection of P. aegyptense from this host order now considered erroneous. Paradiplozoon tetragonopterini (Sterba, 1957) was described from two characid hosts and have not been reported from other host families. Collection of diplozoids from Acipenseriformes, Esociformes and Synbranchiformes are represented by single accounts (either unidentified diplozoids or D. paradoxum) likely from single collections, except for the record from Esox Linnaeus, 1758 (Esocidae; Esociformes) which lists four localities (Khotenovsky, 1985). Collections from the other orders (Cichliformes, Gadiformes, Gobiiformes, Mugiliformes, Perciformes, and Siluriformes) are represented by collections from more than one host, locality, and author, with the exception of Gadiformes for which all records are from Lota lota (Linnaeus, 1758).

Only certain records from Cichliformes, Mugiliformes, Perciformes and Siluriformes specify diplozoid taxa other than D. paradoxum. For Cichliformes, larval P. minutum was reported from O. niloticus, but this was an experimental infection by Paperna (1963). From Mugiliformes, Paradiplozoon kasimii (Rahemo, 1980) was reported from both Planiliza abu (Heckel, 1843) and Planiliza subviridis (Valenciennes, 1836), with Paradiplozoon bliccae (Reichenbach-Klinke, 1961) also reported from P. abu. However, all records of diplozoids from Mugiliformes, except Mhaisen, Al-Rubaie & Al-Sa’adi (2015), are based on unpublished theses (Abdul-Rahman, 1999; Al-Sa’adi, 2007; Al-Janae’e, 2010; Al-Jubori, 2013). Similarly, even though a specific diplozoid, Paradiplozoon pavlovskii (Bychowsky & Nagibina, 1959), was reported from Silurus glanis Linnaeus, 1758, all records of diplozoids from Siluriformes are from unpublished theses (Al-Niaeemi, 1997; Abdul-Rahman, 1999). The highest number of collections of diplozoids from a non-cyprinoid order are from Perciformes, with D. paradoxum or unidentified diplozoids reported from three families and six fish species. Only one case reports on a specific diplozoid species, P. bliccae from Perca fluviatilis Linnaeus, 1758 by Lucký, Navrátil & Jirásková (1989). Most records are for collections in freshwater environments. However, two marine records (Nicoll, 1915; Mhaisen & Al-Maliki, 1996) and three brackish water (lagoon) records (Zaostrovtseva & Evdokimova, 2008; Zaostrovtseva, 2009; Yildirim et al., 2010) were found.

Discussion

Morphometry

The original descriptions of P. ghanense and P. aegyptense contained nearly complete morphometric data and detailed descriptions of the reproductive systems, the latter of which could not be improved upon here. As expected, the data generated from the type series material for both species are similar to that in their original descriptions. Nevertheless, the detailed accounts of haptoral sclerites presented here make a significant contribution to the taxonomic data for both taxa. Additional metric data recorded for P. ghanense included measurements of the hooks, handles, and anoperculum. Regarding the note by Thomas (1957) that no central hooks could be seen, the present authors can attest to the difficulty in studying these structures when not familiar with their appearance. However, the hypothesis by Thomas (1957) that the hooks might have been torn out during the removal of the parasites from the gills is unlikely as the hooks are only used for larval attachment and are entirely enclosed in tissue when the diplozoid matures. The damage to the haptors of the lectotype seen here may have prompted Thomas (1957) to come to this conclusion, but the hooks of the type material could be observed and were studied here. For P. aegyptense, additional metric data generated included the size of the haptor, the space between suckers, and the distance of the suckers to the anterior.

Variation between the data obtained here and the descriptions was mostly seen in slightly wider or narrower ranges of measurements, which are generally negligible and may be attributed to differences in interpretation, calibration, and calculation approaches. For example, the lack of standardized measurement approaches when these taxa were described, which were subsequently provided by Khotenovsky (1985) and later by Pugachev et al. (2010), may also have contributed to the observed variation. This is evident when looking at the clamp sizes of P. ghanense reported by Thomas (1957), where the clamps were longer than wide, opposite to what was recorded here and contrary to the norm for diplozoids. If the length and width of the clamps as given by Thomas (1957) are swopped (see Table 3), they align with the present results. Similarly, the lower range of measurements for both the clamp length and width reported here may be due to Thomas (1957) and Fischthal & Kuntz (1963) not recording all clamps or excluding outliers. Similarly, Fischthal & Kuntz (1963) refer to the anoperculum of the egg of P. aegyptense as the section to which the filament is attached. This is contrary to what is generally accepted for the Diplozoidae, with the operculum considered the smaller section of the eggshell to which the filament is usually attached (Khotenovsky, 1985). As such, the measurement of the opercular end of the egg presented in the description is the anopercular end, which is then highly similar to the data presented here. Printing errors in the original descriptions may also explain some of the differences to the data presented here. For example, the length and the width of the posterior of P. ghanense by Thomas (1957) are identical, which is unlikely. The most notable difference between the data presented here for P. aegyptense and that of Fischthal & Kuntz (1963) is the larger hook sizes, 18.7 (17.7–19.1) here versus 16.5 (16–17). Given the similarity in most other measurements, this discrepancy does not appear to be a calibration error and thus it is possible that the species authorities may have also had some difficultly studying these structures due to their small size, orientation, and familiarity needed to study them. The approaches suggested in Dos Santos, Dzika & Avenant-Oldewage (2019) to study the hooks may offer more conclusive hook morphometry, but the hooks would need to be visualized completely flat at high resolution, which would likely necessitate fresh material and isolation of the sclerites to study them using scanning electron microscopy.

Possibly due to P. ghanense being the only African taxon to infect alestids, morphometry is easily distinguishable from other African taxa. As mentioned, one of the most defining characters of P. ghanense, the lack of division of the anterior clamp jaws, is shared by P. bingolense from Turkey and P. iraqense from Iraq. It would be very illuminating to determine if this similarity between P. ghanense, P. iraqense and P. bingolense is reflected in their phylogenetic relationship, but this requires the genetic study of P. ghanense. Nejat et al. (2023) suggest that P. iraqense may be a junior synonym of P. bingolense as diplozoids collected from the type host of P. iraqense in Iraq, Cyprinion macrostomus Heckel, 1843, show morphological resemblance to the description of P. iraqense, but are genetically identical to P. bingolense. These authors attribute the minor morphological differences between P. bingolense and P. iraqense to the latter species being a morphological variant of P. bingolense. Interestingly, the other species with similar clamp morphology to P. ghanense, P. moroccoense, groups with P. bingolense phylogenetically (Benovics et al., 2021). The presence of an undivided posterior clamp jaw in the holotype of P. aegyptense (see remarks section of P. aegyptense) suggests that this may be a developmental or ancestral state, with the formation of the lateral sclerite absent in less derived taxa or when development is affected by anthropogenic or environmental impacts (Kuperman, 1992; Dzika, 2002; Šebelová, Kuperman & Gelnar, 2002; Dzika, Kuształa & Kuształa, 2007), or natural abnormality frequencies (Zhokhov, Pugacheva & Zhigileva, 2022). This may suggest that P. ghanense and P. bingolense (alongside P. iraqense) possess less “evolved” clamp sclerites and therefore a more ancient morphology. The use of standardised approaches like glycerine ammonium picrate (GAP) (Malmberg, 1957) to study the clamp sclerites of P. ghanense and P. aegyptense would be beneficial for better comparative data, but this again necessitates the collection of fresh material.

The similarity of the clamp sclerite morphology of P. aegyptense and P. krugerense may be linked to both taxa occurring on Labeo species. However, these species are morphometrically distinguishable (see remarks section of P. aegyptense) and there is vast geographical distance between the localities in which P. aegyptense and P. krugerense occur. Additionally, P. vaalense also occurs on Labeo spp., but is distant morphologically and phylogenetically to P. krugerense, irrespective of their geographical proximity (both in South Africa). The study of the genetic profile of P. aegyptense is essential to further comment on the relatedness of the three diplozoids occurring on Labeo spp. in Africa, especially the morphological similarly of P. aegyptense to P. krugerense.

Regarding the voucher for P. ghanense studied here (RMCA_VERMES_35512), the adult metrics were generally smaller than the type material, with especially the clamp sizes more similar to those of P. aegyptense. This may explain why Paperna (1979) identified these parasites as P. aegyptense rather than P. ghanense. However, now that the sclerite detail of the type series for both species have been recorded, alongside the knowledge that the size of the clamps is not a taxonomically reliable attribute (Gläser & Gläser, 1964; Matějusová et al., 2002; Milne & Avenant-Oldewage, 2012), the specimens resemble P. ghanense more closely. This is based on the confinement of the gonads to the fusion area and the detail of the clamp sclerites, especially the lack of lateral additional sclerites connecting the anterior and posterior clamps jaws. The clamps of the P. aegyptense voucher (RMCA_VERMES_35580) are also smaller compared to the type material. However, in this case the reduced size of clamps may be attributed to different host sizes or species, with the type material collected from L. forskalii and the voucher from Labeo coubie Rüppell, 1832. Locality or the season when it was collected may also play a role as the type material was collected in Egypt (presumably the Nile), while the P. aegyptense voucher was collected from the Lake Volta in Ghana. Similarly, the voucher material of P. ghanense was collected in Uganda while the types were collected in Ghana, and in this case the host species could not play a role in the variation as the voucher material was collected from the type host. Differences in preparation techniques may have played a role in the variation between the type series and the vouchers. Thomas (1957) used a corrosive acetate, Ehrlich’s acid haematoxylin, and Mayer’s carmalum, while the preparation method is not provided by Paperna (1979). Paperna (1979) likely used “Semicohn Carmin” (Semichon’s carmine) as in Paperna (1963). Similarly, Fischthal & Kuntz (1963) killed the parasites in hot water, fixed them overnight in alcohol-formalin-acetic acid (AFA or FAA), washed and stored them in 70% ethanol with 1–2% glycerine, stained them using Harris haematoxylin and Mayer’s paracarmine, and mounted them in balsam. Again, such detail was not provided in Paperna (1969), but also likely involved Semichon’s carmine, as in Paperna (1963).

The collection of the voucher specimen for P. aegyptense from L. coubie alongside the morphometric similarity of the voucher to the type series material may indicate that the unidentified diplozoid collected from the same host by Sicard et al. (2003) may represent P. aegyptense as hypothesised in Dos Santos & Avenant-Oldewage (2020). The rDNA data for the specimens studied by Sicard et al. (2003) were “strictly similar” (sic.) to P. homoion, prompting Dos Santos & Avenant-Oldewage (2020) to form three hypotheses for this similarity: P. aegyptense is a junior synonym of P. homoion; there was a mix-up with the data in Sicard et al. (2003), or that P. homoion infects indigenous African hosts as well. When comparing the clamp sclerites and testes shape of P. homoion and P. aegyptense, their synonymy is highly unlikely. The smaller hooks of P. aegyptense further distance it from P. homoion. Thus, only the latter two explanations seem plausible. Unfortunately, without obtaining haplotypes from morphometrically confirmed specimens of P. aegyptense, further speculations on the topic would be unsupported. Paradiplozoon aegyptense has also been recorded from native hosts in India (Ahmad et al., 2015a, 2015b, 2015c). The morphometry for P. aegyptense from Schizopyge niger (Heckel, 1838), in India recorded by Ahmad et al. (2015a) is similar, but generally smaller than the data for the types. However, the data provided by Ahmad et al. (2015a) lack completeness and omit taxonomically essential information. It is also unclear whether the morphometry for P. aegyptense presented in Ahmad et al. (2015a) represents their own results or a repetition of data from Fayaz & Chishti (1999) that was mis-cited with the incorrect date (1993) as the authority of P. aegyptense. Furthermore, the images provided in Ahmad et al. (2015a) are not clear, but the general silhouette is not similar to that of P. aegyptense. Thus, the taxonomic identity of the specimens collected in India need to be revised as the diplozoids in question do not represent P. aegyptense, but constitute a diplozoid taxon not previously genetically characterised (Dos Santos & Avenant-Oldewage, 2020). Following the idea that diplozoid phylogeny may reflect clamp sclerite characteristics by Nejat et al. (2023), P. ghanense and P. aegyptense may be phylogenetically close to P. bingolense and P. krugerense, respectively, due to the similarity of their clamp sclerites.

Host specificity in Africa

As noted, the details for RMCA_VERMES_35512 match the details reported in Paperna (1979) for P. aegyptense collected from B. macrolepidotus in Uganda. Interestingly, Paperna (1979) does not provide the RMCA accession number for this specimen (M.T. 35.512 historically), but he provides an accession number for P. aegyptense collected from L. coubie (M.T. 35.580, studied here), which likely relates to Paperna (1969). Nevertheless, the re-identification of RMCA_VERMES_35512 as P. ghanense markedly simplifies the host specificity of Paradiplozoon spp. in Africa, with P. ghanense exclusively from alestids, P. aegyptense from cyprinoids, and all other Paradiplozoon spp. from cyprinids. Thurston (1970) notes that the Diplozoon sp. from Alestes sp. could not be identified due to suboptimal fixation, but that it resembled P. ghanense, matching what would be expected from this host. Most African Paradiplozoon taxa occurring on more than one host species appear to be restricted to hosts of the same genus, or at least family. For example, P. vaalense and P. krugerense occur only on Labeo spp., while P. ichthyoxanthon occurs on Labeobarbus species. Paradiplozoon ghanense infects two host genera, Brycinus Valenciennes, 1850 and Alestes Müller & Troschel, 1844. However, B. macrolepidostus and Alestes baremoze (Joannis, 1835) were congeners in the past, indicating their taxonomic relatedness. Similarly, the unidentified Paradiplozoon sp. collected in South Africa by Truter et al. (2023) occurs on two cyprinids from different genera, Cheilobarbus serra (Peters, 1864) and Sedercypris calidus (Barnard, 1938). These hosts were congeners in the past and are still in the same subfamily (Smiliogastrinae), suggesting a close taxonomic relationship. Unfortunately, Truter et al. (2023) did not identify the Paradiplozoon sp. they collected, but the clamp sclerite seem to resemble those of P. ichthyoxanthon.

The host specificity for P. aegyptense is more complex as it was described from a Labeo sp. and subsequently recorded from three additional Labeo spp. (plus un unidentified Labeo), one Enteromius sp. and one Raiamas Jordan, 1919 sp. in Africa (see Table 1 for details). Dar et al. (2012a, 2012b) and Ahmad et al. (2015a, 2015b, 2015c) also recorded P. aegyptense from non-African hosts in India. These host species belong to four different genera, Cyprinus Linnaeus, 1758, Schizopyge Heckel, 1847, Schizothorax Heckel, 1838 and Carassius Jarocki, 1822. However, as discussed here and in Dos Santos & Avenant-Oldewage (2020), these records of P. aegyptense from India are highly unlikely and are considered invalid until morphologically substantiated. Nevertheless, even if these records are considered valid, all the fishes from which P. aegyptense have been reported are Cypriniformes. Its collection from a single Danionidae species may be indicative of an accidental infection or distinct taxon as all other hosts are cyprinids, and needs re-investigation.

Host specificity of African Paradiplozoon taxa is also illustrated by negative data presented by some authors. Thomas (1957) notes that although several representatives of many families were investigated, only B. macrolepidotus were infected with diplozoid parasites. The authors mention that members of the Cyprinidae, Mormyridae, Citharinidae, Bagridae, Schilbeidae, Clariidae, and Mochokidae were examined for diplozoid infections, but not the specific species collected. Similarly, both P. vaalense and P. ichthyoxanthon occur in the same system on different host genera, with no crossover reported or any record of any other fishes infected with diplozoids in the system (Avenant-Oldewage et al., 2014; Avenant-Oldewage & Milne, 2014; Dos Santos, Jansen van Vuuren & Avenant-Oldewage, 2015; Dos Santos & Avenant-Oldewage, 2015). Paradiplozoon krugerense shows similar host specificity as various species of Cyprinidae (other than the two Labeo type hosts), Characidae, Cichlidae, Clariidae, Schilbeidae and Mochokidae sampled within the Olifants and Selati Rivers were not infected by diplozoids (Dos Santos & Avenant-Oldewage, 2016). The unidentified Paradiplozoon sp. also only occurred on C. serra and S. calidus, while three other cyprinids collected during the same survey were not infected (Truter et al., 2023). Interestingly, Truter et al. (2023) also collected ten specimens of a Labeobarbus sp. (host genus for P. ichthyoxanthon), but these were not infected with diplozoids. The diplozoids from C. serra and S. calidus may thus represent a distinct taxon based on the negative Labeobarbus data, but a species closely related to P. ichthyoxanthon based on the clamp sclerite similarity.

Diplozoids infecting non-cyprinoids

As can be seen from Table 5, well documented reports of diplozoids from non-cyprinoids is exceedingly rare. The only diplozoids not described from Cypriniformes are P. tetragonopterini from the characids Ctenobrycon spilurus (Valenciennes, 1850) and Gymnocorymbus ternetzi (Boulenger, 1895) in the Erfurter Aquarium in Germany, and P. ghanense. The subsequent collections of P. ghanense from other alestids and from different localities substantiated the occurrence of this species on the hosts, while Sterba (1957) was able to infect the type host species with oncomiracidia of P. tetragonopterini experimentally. Sterba (1957) also infected another characid, Psalidodon anisitsi (Eigenmann, 1907), with lesser success, even noting C. spilurus as the preferred host over G. ternetzi. Several other tetra species, and even cyprinids, could not be infected by Sterba (1957), indicating specificity for characids. Although P. tetragonopterini appears to infect South American characids, this taxon has not been reported again and the origin of the species is still unknown as the hosts were collected in aquarium tanks with several other fishes from various origins. Paradiplozoon tetragonopterini is also unique among diplozoids in the morphology of its testis, as well as having an anopercular spine on the egg which is very uncommon for the genus. This may indicate a more distant relation to other diplozoid taxa, possibly related to the suspected geographical origin of the species, being the only South American diplozoid. Unfortunately, the clamp sclerites of P. tetragonopterini have not been studied conclusively, and thus it is not known if it shares the most distinguishing feature of P. ghanense in comparison to most diplozoid taxa—the absence of a lateral sclerite. It would be very interesting to study the genetic profile of the two taxa occurring on Characiformes to determine if they are more closely related than to other diplozoids. The presence of a single diplozoid taxon infecting Characiformes in Africa and the Americas (the only regions in which these fishes occur) may also be very informative from an evolutionary timescale perspective, possibly allowing for the calibration of a molecular clock for the family.

The remainder of records for diplozoids occurring on non-cyprinoids have very little information as they are part of lists or reviews (Khotenovsky, 1985; Aioanei, 1996), general surveys (Nicoll, 1915, 1924; Paperna, 1969; Thurston, 1970; Paperna, 1979; Batra, 1984; Lucký, Navrátil & Jirásková, 1989; Mhaisen & Al-Maliki, 1996; Echi & Ezenwaji, 2010; Yildirim et al., 2010; Zaostrovtseva & Evdokimova, 2008; Zaostrovtseva, 2009; Mhaisen, Al-Rubaie & Al-Sa’adi, 2015) or unpublished theses (Paperna, 1963; Al-Niaeemi, 1997; Abdul-Rahman, 1999; Al-Sa’adi, 2007; Al-Janae’e, 2010; Al-Jubori, 2013). Khotenovsky (1985) provides the most extensive list of diplozoid collections, including those from non-cyprinoid hosts. However, it is not clear if the information therein is from personal collections or summaries of other sources (like Nicoll (1915, 1924), Paperna (1963), Thurston (1970), and Batra (1984)) as no specific collection details or citations are given. However, all non-cyprinoid reports in Khotenovsky (1985) have published complimentary data, either as the likely source of the record or a subsequent collection from the same host. The only exceptions are the collection of unidentified diplozoids from Esox lucius Linnaeus, 1758, Neogobius melanostomus (Pallas, 1814), Gymnocephalus cernua (Linnaeus, 1758), Sander lucioperca (Linnaeus, 1758), and Sander volgensis (Gmelin, 1789) which were only reported by Khotenovsky (1985). However, diplozoids have been collected from other hosts in the same families as the latter four exceptions, with only the report from Esocidae (Esociformes) not having a familial counterpart.

The collection of D. paradoxum from Gasterosteus aculeatus Linnaeus, 1758, a marine species, by Nicoll (1915) (and later Khotenovsky (1985)), is surprising as species of Diplozoidae are generally considered exclusively freshwater parasites. However, Table 5 contains other accounts from euryhaline or brackish water fishes (e.g., P. fluviatilis and Neogobius melanostomus (Pallas, 1814)), and even catadromous (Anguilla anguilla (Linnaeus, 1758)) or anadromous (Acipenser gueldenstaedtii Brandt & Ratzeburg, 1833) fishes from lagoons or deltas. The repetitive collection of diplozoids from certain species also indicate that these may not be merely accidental infections. For example, the collection of diplozoids (D. paradoxum, P. bliccae and unidentified diplozoids) from P. fluviatilis has been noted on at least three separate occasions in different localities (Khotenovsky, 1985; Lucký, Navrátil & Jirásková, 1989; Aioanei, 1996; Zaostrovtseva & Evdokimova, 2008; Zaostrovtseva, 2009). Additionally, the prevalence of D. paradoxum on P. fluviatilis recorded by Zaostrovtseva & Evdokimova (2008), was relatively high (47.8–50%, 20–37 fish per site) and thus unlikely an accidental infection as up to 50 parasites were collected from some hosts. This occurrence of diplozoids on Perciformes appears generally accepted, even though this topic has yet to be confirmed using morphological or molecular tools. The repetitive collection of diplozoids from P. abu also needs further attention, but only one of the five records detailing the collection of diplozoids from this host is a published, peer-reviewed account (Mhaisen, Al-Rubaie & Al-Sa’adi, 2015). The presence of D. paradoxum on Coptodon zillii (Gervais, 1848) in Turkey (Yildirim et al., 2010) is one of the only published records of a diplozoid from a non-cyprinoid accompanied by an image of the parasite, but the low quality of the image hinders identification of the specimen, and the prevalence was low (4.17%.). In contrast, records such as the collection of D. paradoxum from A. anguila by Zaostrovtseva & Evdokimova (2008) showed exceptionally high infections (86.7% prevalence on 64 fish) and warrant further investigation. Reports on the presence of diporpa, like Mhaisen, Ali & Khamees (2013), may be purely accidental as they usually do not survive long on the incorrect host and thus rarely fuse and mature (Paperna, 1963). Thus, the proper morphological and genetic study, alongside noting the developmental stage, of diplozoids from non-cyprinoid hosts would be greatly beneficial and would most likely uncover new taxa.

Conclusion

The present study not only completes the morphometric data for P. ghanense and P. aegyptense to allow comprehensive taxonomic comparison with other diplozoids, but also designated the type series for P. ghanense and confirms the paratype status of six P. aegyptense specimens. Additionally, the re-identification of a P. ghanense specimen has allowed for the partial resolution of discrepancies in the host specificity of this genus in Africa. The next step would be to obtain genetic material for these species to allow for molecular analyses to confirm their taxonomic distinctness and their phylogenetic relations. The more diligent study of diplozoids from non-cyprinoid hosts also appears to be a promising topic for future studies, likely resulting in the description of several diplozoid species.

Supplemental Information

Supplemental Information 1 Raw measurement data for Paradiplozoon ghanense (Thomas, 1957) specimens studied.

Supplemental Information 2 Raw measurement data for Paradiplozoon aegyptense (Fischthal & Kuntz, 1963) specimens studied.

Supplemental Information 3 Illustration of general diplozoid clamp sclerites.

Supplemental Information 4 Photomicrograph of Paradiplozoon ghanense(Thomas, 1957) holotype USNM 1367213 from Brycinus macrolepidotus Valenciennes, 1850.

Supplemental Information 5 Photomicrographs of the dorsal joining sclerite of Paradiplozoon ghanense(Thomas, 1957) and Paradiplozoon aegyptense (Fischthal & Kuntz, 1963) indicated by arrows.

Supplemental Information 6 Illustrations of the haptoral sclerites of Paradiplozoon ghanense (Thomas, 1957) voucher RMCA_VERMES_35512 from Brycinus macrolepidotus Valenciennes, 1850.

Supplemental Information 7 Photomicrographs of Paradiplozoon ghanense (Thomas, 1957) voucher RMCA_VERMES_35512 from Brycinus macrolepidotus Valenciennes, 1850.

Supplemental Information 8 Photomicrograph of Paradiplozoon aegyptense (Fischthal & Kuntz, 1963) voucher RMCA_VERMES_35580 from Labeo coubie Rüppell, 1832.

The authors would like to thank Dr. Tine Huyse (RMCA) for lending the material studied here to the authors. The authors also thank Dr. Anna Philips for facilitating the study of the material at the SNMNH by QMDS. Finally, Mr. Chad Walter is thanked for generously hosting QMDS during his visit to the SNMNH. The authors thank the central analytical facility (Spectrum) and the Faculty of Science at the UJ for access to infrastructure and equipment.

Additional Information and Declarations

Competing Interests

Author Contributions

Data Availability

Annemarie Avenant-Oldewage is an Academic Editor for PeerJ.

Quinton Marco Dos Santos conceived and designed the experiments, performed the experiments, analyzed the data, prepared figures and/or tables, authored or reviewed drafts of the article, and approved the final draft.

Annemariè Avenant-Oldewage conceived and designed the experiments, performed the experiments, analyzed the data, authored or reviewed drafts of the article, funding acquisition and supervision of PhD and PDRF, and approved the final draft.

The following information was supplied regarding data availability:

The raw data are available in the Supplemental Tables.

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
