# Peer review of "Revisiting the type material of two African Diplozoinae (Diplozoidae: Monogenea), with remarks on morphology, systematics and diplozoid specificity"

_PeerJ, doi:10.7717/peerj.17020_

## Round 0.1 · original submission · Major Revisions

I believe that this work will be useful because you examined type material from several collections.

Both reviewers suggested major revision. I agree with them that the text need to be completely changed in the revised manuscript.

The abstract should also be completely rewritten - the current version does not correspond to what you expect from an abstract.

An important point of zoological nomenclature. I urge you to read Article 73.1.3. of the code:
"73.1.3. The holotype of a new nominal species-group taxon can only be fixed in the original publication and by the original author (for consequences following a misuse of the term "holotype" see Article 74.6)."

In your revised version, please refer to appropriate articles of the Code if you make any nomenclatural decisions, and cite the Code in the list of references.

Reviewer 1 ·

Basic reporting

The MS brings supplementary information to the original description of two Paradiplozoon species (a definitively important contribution and it is appreciated), but not the revision.

The title of the MS does not correspond with the main results presented in the MS. There is no part on host specificity in your results.

The abstract should represent the summary of the most important outputs of this work, not the case of the current version.

The MS requires a substantial rearrangement in order to reach high quality.

The English would need professional proofreading as the formulation of the current version lowers its quality.

The introduction does not represent a strong introduction into the context of the issue, it is quite fragmented.

You did a great search on available literature as can be seen from Table 4, why table form does not form part of your results?

You spent plenty of time collecting the literature and compiling tables but you did not make proper use of it.

Discussion is unnecessarily too long.

Table 3 should actually be Table 1 and the most important information should be extracted in the introduction, not the case of the current version.

Experimental design

There is no experimental design to be applied to the current MS.

The section on Material and methods requires substantial improvement

Validity of the findings

A better formulation of the most important outputs of the MS should be applied.

Conclusions need to be rewritten.

Additional comments

I have included many comments directly in the PDF.

Annotated reviews are not available for download in order to protect the identity of reviewers who chose to remain anonymous.

Reviewer 2 ·

Basic reporting

The authors have provided detailed morphological redescriptions of two African diplozoids, namely P. ghanense and P. aegyptense, based on the available specimens. They have effectively elucidated the taxonomically significant structures, particularly the haptoral sclerites, and have furnished comprehensive morphometric data. Additionally, the authors have made a compilation of host and locality records concerning African diplozoids, enriching our understanding of these parasites. I recommend accepting the paper for publication after considering the comments and recommendations.

There is room for improvement in the clarity and conciseness of the English language used in the discussion section. Certain sentences appear casual and verbose, which might hinder understanding for an international audience. I have marked some areas in the article where I have difficulty understanding, and have made a few suggested changes.
The “Introduction” section lacks logical coherence. I suggest that you firstly provide an introduction to the Diplozoidae Palombi, 1949, emphasizing its taxonomic significant features and host specificity, then make a concise sumarization of taxonomy of African diplozoid to establish the existing knowledge base, and finally describe the taxonomic situation of P. ghanense and P. aegyptense in detail and state the weakness of original descriptions.
I strongly recommend that the authors provide well-labeled drawings of an entire clamp, highlighting the ventral and dorsal positions of each part.

Experimental design

Among the seven diplozoid species described from Africa, the original descriptions of both P. ghanense and P. aegyptense lack key taxonomic characteristics. To add the essential morphological details, the authors checked All speciesment available using standard compound microscopy techniques, including phase contrast and differential interference contrast (DIC) microscopy, and drawed clamp and hook faithfull.
The terminology used for the clamp sclerites should be clear. Typically, diplozoid clamp sclerites consist of the anterior clamp jaw, posterior clamp jaw, median sclerite, trapeze spur, anterior joining sclerite and posterior joining sclerite (Khotenovsky, 1985; Benovics et al., 2021). However, the authors refer to many clamp sclerites as “additional sclerite" in the description of P. aegyptense, which can potentially confuse readers. To avoid confusion, you can either follow the nomenclature used by Khotenovsky (1985) and Benovics et al. (2021), or make your modifications while clearly labeling them in the figures.

Validity of the findings

The authors have effectively elucidated the taxonomically significant structures, particularly the haptoral sclerites, and have furnished comprehensive morphometric data. However, much of the details they described in their TEXT, is not clearly shown in their figures. Given the complexity of the clamp structure as showed in the figure, separate figures for each clamp sclerite should be provided for clarity.
I commend the authors for providing detailed tracable information about the meterial. I noted that the authors redesignated a voucher (RMCA_VERMES_35512) as P. ghanense instead of P. aegyptense, which was originally named by Paperna. While the author has described the diporpa separately and mentioned that the morphology of the adult in the voucher closely resembles that of the type material, no pictures were provided. I appreciate your valuable input, but providing supplementary visual
evidence would further strengthen the validity of your findings.
it is much better to provide well-resolution of photograph of Clamps from different material as the supplementary.

Additional comments

In the“Discussion” section, the authors primarily discussed the differences in morphometric characteristics. It would be beneficial for the authors to provide further elaboration on the anatomical features, specifically addressing the knowledge gap that their study aims to fill.
Most of the other comments placed on the manuscript are relatively minor but should be seriously considered before acceptance

Annotated reviews are not available for download in order to protect the identity of reviewers who chose to remain anonymous.

---

## Round 0.2 · Minor Revisions

I apologize for the delay. One reviewer was late, then I took some days away from work.

1. Lectotype. I agree that your decision to designate a lectotype and paralectotype is certainly better than your original interpretation;

2. Title. One reviewer has commented the title. I believe you can keep Monogenea, which will remain an important keyword even when most researchers are convinced that the Monogenea are not monophyletic, as I am.
However, the title is still long. Perhaps you should consider something shorter, like:
“The type material of two African Diplozoidae (Monogenea), with remarks on morphology, systematics and specificity.” This is just a suggestion.

3. Abstract. In the abstract, it seems acceptable to me to provide the authors of the specific taxa but not the authors of the upper-level taxa.

4. Other.
In the text, it seems acceptable to me to omit the authors of all supraspecific fish taxa.
Line 508 – I agree with the reviewer that this probably never happens. Studies of old specimens kept in Canada balsam never reported such a phenomenon.
The discussion can probably be shortened. Please re-read all the text – both reviewers were right in reporting many misspelled words and other errors. This should not happen.

Reviewer 1 ·

Basic reporting

The current version of the MS represents a substantial improvement since its first version.

The title of the MS is bit odd, specially the second part.

I suggest a bit of re-arrangement of the introduction section to have a better flow.

The major issues are the inconsistency in the citing of the authorities, discussing points which don' t form a part of the MS's results.

Some of used formulation in simplified version as of Diplozoinae is not he optimal and correct.

I have indicated issues and changes directly into a PDF

Experimental design

It would be beneficial to combine aims of the MS into one section and formulate them more directly.

Validity of the findings

Authors manage to gather various literature source on the topic of diplozoid parasites in Africa, what have an additional values for other researcher working in the field.

Annotated reviews are not available for download in order to protect the identity of reviewers who chose to remain anonymous.

Reviewer 2 ·

Basic reporting

The author significantly improved the article by strengthening logical coherence, and creating an organized structure that greatly improves its readability.
There are still some very minor language spelling errors or unclear parts in the article
-line 150:" the of"
-line 172: "there" should be "their"
-line 293:" A such" should be " As such"
-line 385: "similarly" should be "similarity"
-line 401: "The morphology of"
-line 491-492: "the detailed accounts of haptoral sclerites presented here make a significant contribution to the taxonomic data for both taxa."
-line 459: "with" should be "while"
-line 702: what diplozoids is "these diplozoids"
-line 708:"from Cypriniformes"? but both the alestids and characids is of Characiformes as shown in your table 5.

Experimental design

no comment

Validity of the findings

The authors thoroughly examined all available specimens and provided well-labeled drawings of clamp sclerites for two diplozoids that lacked essential taxonomic information in the original descriptions. Additionally, they compiled complete host and locality records for African diplozoids. I believe that this work is valuable for developing a more comprehensive understanding of this group in Africa.

---

## Round 0.3 · accepted · Accept

Please note that you title is not exactly the one I suggested. In my opinion, the addition of "their" is confusing here.